# Study on the Effect of Pharmaceutical Excipient PEG400 on the Pharmacokinetics of Baicalin in Cells Based on MRP2, MRP3, and BCRP Efflux Transporters

**DOI:** 10.3390/pharmaceutics16060731

**Published:** 2024-05-29

**Authors:** Dan Yang, Min Zhang, Mei Zhao, Chaoji Li, Leyuan Shang, Shuo Zhang, Pengjiao Wang, Xiuli Gao

**Affiliations:** 1State Key Laboratory of Functions and Applications of Medicinal Plants, School of Pharmacy, Guizhou Medical University, Guiyang 550025, China; 18311824619@163.com (D.Y.); minzhang@gmc.edu.cn (M.Z.); 17385484160@163.com (L.S.); wangpengjiao@gmc.edu.cn (P.W.); 2Center of Microbiology and Biochemical Pharmaceutical Engineering, Department of Education of Guizhou, Guiyang 550025, China; 3School of Basic Medical Sciences, Guizhou Medical University, Guiyang 550025, China; 4Experimental Animal Center, Guizhou Medical University, Guiyang 550025, China

**Keywords:** PEG400, cellular pharmacokinetics, baicalin, transporters, vesicles

## Abstract

Pharmaceutical excipient PEG400 is a common component of traditional Chinese medicine compound preparations. Studies have demonstrated that pharmaceutical excipients can directly or indirectly influence the disposition process of active drugs in vivo, thereby affecting the bioavailability of drugs. In order to reveal the pharmacokinetic effect of PEG400 on baicalin in hepatocytes and its mechanism, the present study first started with the effect of PEG400 on the metabolic disposition of baicalin at the hepatocyte level, and then the effect of PEG400 on the protein expression of baicalin-related transporters (BCRP, MRP2, and MRP3) was investigated by using western blot; the effect of MDCKII-BCRP, MDCKII-BCRP, MRP2, and MRP3 was investigated by using MDCKII-BCRP, MDCKII-MRP2, and MDCKII-MRP3 cell monolayer models, and membrane vesicles overexpressing specific transporter proteins (BCRP, MRP2, and MRP3), combined with the exocytosis of transporter-specific inhibitors, were used to study the effects of PEG400 on the transporters in order to explore the possible mechanisms of its action. The results demonstrated that PEG400 significantly influenced the concentration of baicalin in hepatocytes, and the AUC_0–t_ of baicalin increased from 75.96 ± 2.57 μg·h/mL to 106.94 ± 2.22 μg·h/mL, 111.97 ± 3.98 μg·h/mL, and 130.42 ± 5.26 μg·h/mL (*p* ˂ 0.05). Furthermore, the efflux rate of baicalin was significantly reduced in the vesicular transport assay and the MDCKII cell model transport assay, which indicated that PEG400 had a significant inhibitory effect on the corresponding transporters. In conclusion, PEG400 can improve the bioavailability of baicalin to some extent by affecting the efflux transporters and thus the metabolic disposition of baicalin in the liver.

## 1. Introduction

Pharmaceutical excipients refer to excipients and additives used in the production of medicines and in the formulation of prescriptions and are substances other than active ingredients or prerequisites that have been reasonably evaluated in terms of safety and are generally included in pharmaceutical preparations. With the understanding of the process of drug absorption and metabolism in the body, an increasing number of studies have demonstrated that pharmaceutical excipients have the potential to affect the function of drug gastrointestinal transporters, affecting the process of absorption, distribution, metabolism, and excretion of drugs and thus affecting drug bioavailability. Therefore, focusing on the interaction between excipients and active drugs is of current relevance [1,2,3]. Polyethylene glycol 400 (PEG400) is a mixture of ethylene oxide and water by polycondensation. It is a common medicinal excipient in traditional Chinese medicine *Scutellaria baicalensis* Georgi-related compound preparations, such as Yinzhihuang soft capsules and Yinhuang Hanhua dropping pills. Previous studies have indicated that the use of PEG400 can enhance the bioavailability of the drug ranitidine in men [4]. The study of whether and how PEG400 will affect the metabolic disposal of active components of traditional Chinese medicine in vivo is necessary to clarify the mechanism of the interaction between excipients and drugs (i.e., drug interaction, DDI). It is also one of the hotspots of basic research in the rational use of traditional Chinese medicine preparations, biopharmaceutics, and excipients, as well as clinical rational drug use.

*Scutellaria baicalensis* is the dried root of the plant *Scutellaria baicalensis*, which belongs to the family Labiatae. It has a long history of medicinal use and is rich in medicinal resources, with the effects of clearing heat and drying dampness, diarrhea, and detoxification. Baicalin (BG) is one of the principal active ingredients of *Scutellaria baicalensis* and has a wide range of pharmacological activities, including anti-inflammatory, antibacterial, antioxidant, and cardioprotective effects [5,6,7,8]. Although baicalin exhibits a wide range of pharmacological activities, it is a typical flavonoid with poor water and lipid solubility, which renders it difficult to be absorbed through the small intestinal epithelium by passive diffusion, leading to malabsorption and reduced tissue distribution, and the oral bioavailability of typical flavonoids is very poor. Consequently, a more detailed understanding of their in vivo behavior will be crucial for the rational clinical application of these agents, which is of great significance for targeted drug delivery and precision medicine.

Drug transporters are a class of transmembrane proteins that are widely distributed in the human body and mediate the entry and exit of endogenous and exogenous substances into and out of biological membranes, which are mainly categorized into ATP-binding cassette (ABC) transporters and solute carrier (SLC) transporters [9]. ABC transporters are a family of transporters that have been extensively studied, including three types of transporter proteins: P-glycoprotein (P-gp), the breast cancer resistance protein (BCRP), and the multidrug resistance-associated protein (MRP), which mediate the transport of drugs and endogenous substances by utilizing the energy from the hydrolysis of ATP [10]. Alterations in the expression and function of these three types of proteins frequently result in pharmacokinetic changes and are major targets for drug interactions [11]. Previous studies have shown that after oral administration of baicalin (BG), baicalin (BG) is converted to baicalein (B) across the membrane into the small intestinal mucosa under the action of glucuronidase of intestinal flora; subsequently, BG and baicalein-6-O-β-D-glucuronoside (B6G, isomer of baicalin) are catalyzed by intestinal phase II metabolizing enzyme UGTs. Phase II metabolite baicalin (BG) and its metabolic isomer B6G are the main metabolites of oral baicalein in blood [12,13]. These phase II metabolites are simultaneously excreted to the intestinal lumen or body circulation by ATP-binding cassette protein transporters (ABCs) (P-gp, BCRP, MRPs, etc.), which are located in the apical membrane of intestinal epithelial cells, and BG and B6G in the body can be transported back to the intestinal lumen via the bile and intestinal mucosa to undergo reabsorption processes [14,15,16]. Previous studies have shown that pharmaceutical excipients such as PEG400 after gavage in rats were used to determine the level of P-gp protein expression in different intestinal segments using the western blot technique, of which the results indicated that PEG400 significantly reduced the protein expression of the drug transporter P-gp in the intestine [17]. Pharmaceutical excipients such as Cremophor EL and PEG400 have been demonstrated to inhibit MRP2, thereby increasing the bioavailability of drugs that act as MRP2 substrates [18]. Consequently, metabolizing enzymes and transporters represent the primary determinants of drug pharmacokinetics in vivo, and changes in their expression and function cause changes in pharmacokinetics and are the main targets for prior drug interactions [11].

Intracellular pharmacokinetics is the study of the kinetic process of absorption, transport, distribution, metabolism, and excretion of drugs in cells and subcells as a microscopic whole [19]. An increasing amount of research has found that classical pharmacokinetics cannot fully explain the pharmacological effects of drugs in specific tissues or cells, so it is challenging to accurately predict the efficacy of drugs in vivo. Generally speaking, the drug effect is through multiple biological barriers to reach the cell and interact with many specific targets in the cell, which in turn produce the related biological effect [20]. The drug concentration around the intracellular target can more truly reflect the drug effect than the plasma drug concentration [21]. A significant number of drug design processes now refer to the study of cell pharmacokinetics of traditional Chinese medicine to find the main factors that affect the clinical efficacy and, according to these strategies, to improve the cell pharmacokinetic behavior in order to increase the effect of drugs [22,23,24,25].

In vitro analysis of ABC transporters employs two fundamental approaches: the use of intact cells expressing the transporter or the utilization of membrane vesicles overexpressing the studied transporter protein [18]. In recent years, numerous researchers at home and abroad have used cell, gene knockout, and gene transfection cell or animal models to study the single action and coupling between intestinal drug metabolic enzymes and efflux transporters [26]. The vesicular transporter assay was prepared from transgenic Sf9 cells overexpressing a specific ABC transporter, which can be used to determine the effect of drugs on the transporter by detecting changes in the concentration of the substrate drug in the extravascular ABC transporter [27].

Therefore, in the present study, the effect of PEG400 on the metabolic kinetics of BG at the hepatocyte level was investigated using human-derived hepatocellular carcinoma (HepG2) cells [28], and the changes in the content of baicalin and its metabolites were determined in HepG2 cells. The western blot and other techniques were employed to investigate the effect of protein expression of the major transporters of baicalin (BCRP, MRP2, and MRP3) and thus its possible mechanism of action. The effects of PEG400 on MRP2, MRP3, and BCRP efflux transporters were investigated by vesicular transport experiments of MRP2, MRP3, and BCRP and cell models of MDCKII-WT, MDCKII-MRP2, MDCKII-MRP3, and MDCKII-BCRP. The study on the effect of PEG400 on the pharmacokinetics of baicalin in hepatocytes and its possible mechanism will help to clarify the strategy to improve the oral bioavailability of baicalin and provide a scientific basis for the scientific use, further development, and clinical rational use of excipients of traditional Chinese medicine preparations.

## 2. Materials and Methods

### 2.1. Materials

Baicalein-7-O-β-D-glucuronide (BG), baicalein, and genistein (IS) standards (purity ≥ 98%) were provided by Yuanye Bio (Shanghai, China). Baicalein-6-O-β-D-glucuronide (B6G) standard (purity ≥ 95%) was purchased from Dalian Medical University. PEG400, Hank’s buffered saline solution (HBSS), phosphate-buffered saline (PBS), Tris-HCl buffer (PH = 7.4), and BCA protein quantification kit were purchased from Solarbio (Beijing Solarbio Biotechnology Co., Ltd., Beijing, China) The following reagents were purchased from MCE (MedChemExpress), Monmouth Junction, NJ, USA: MK-571 hydrate sodium salt (MK-571), Ko-143 hydrate (Ko143), and indomethacin. The human BCRP, MRP3, and MRP2 vesicles were purchased from Shanghai Ruide Liver, Shanghai, China. HPLC grade acetonitrile and methanol were produced by Merck (Darmstadt, Germany). Transwell plates (12-well, 0.4 μm polycarbonate membrane) were purchased from Jet Bio, Guangzhou, China. Filter membranes (0.7 μm glass fiber membranes) used to detect vesicle transit were purchased from Pall Corporation, New York, NY, USA. The Millicell^®^-ERS-2 resistivity meter was purchased from Merck & Co, Darmstadt, Germany.

### 2.2. Methods

#### 2.2.1. Cell Culture

The HepG2 cells were purchased from the Kunming Cell Bank of the Typical Cultures Preservation Committee of the Chinese Academy of Sciences. MDCK II-WT and MDCK II-BCRP cells were from the generous gift of Prof. Xuehua Jiang (College of Pharmacy, Sichuan University, Chengdu, China), MDCK II-MRP2 was a gift from Prof. Xiaoyan Chen of the Shanghai Institute of Pharmaceutical Sciences, and MDCK II-MRP3 cells were commissioned to be constructed by Yuanjing Biologicals in Guangzhou, China. Western blot analysis demonstrated the overexpression of each protein. The cells were cultured in Dulbecco’s Modified Eagle Medium (DMEM) containing 10% fetal bovine serum, 100 IU/mL penicillin, and 100 μg/mL streptomycin at pH 7.4, temperature 37 °C, and CO_2_ concentration of 5%. The cells were passaged with 0.25% trypsin-EDTA solution at 80–90% confluence. All cells used in this study were between generations 10 and 20 [29].

#### 2.2.2. UPLC-MS/MS Analysis of Sample Concentration

The samples were quantified by UPLC-MS/MS, which consisted of a Dionex Ultimate 3000 UPLC system (Thermo Fisher Scientific, Waltham, MA, USA) and a triple quadrupole tandem mass spectrometer (Thermo Fisher Scientific, Waltham, MA, USA). The mass spectrometry was performed using an ESI ion source and in positive ion mode. The chromatograms are shown in Appendix A. The samples were analyzed by a Waters XBridge BEH C_18_ column (2.1 × 100 mm). Mobile phase A consists of acetonitrile, while mobile phase B consists of water containing 0.1% formic acid. The elution program was as follows: B was initially 90%, 0 to 9 min B decreased from 90% to 27%, and 9.1 min to 12 min increased from 27% to 90% B. The flow rate of the mobile phase was 0.3 mL/min, the temperature of the automatic injector was 5 °C, and the injection volume was 10 μL. The *m*/*z* for BG and B6G was 447 → 271, for baicalein *m*/*z* 271 → 123, and for genistein (IS) *m*/*z* 271 → 153. Other mass spectrometry parameters included sheath gas (Arb) pressure, 20 psi, and auxiliary gas (Arb) pressure, 2 psi; source voltage, 4.8 KV, and capillary voltage, 2.5 KV; and ion-transfer-tube temperature, 375 °C, and atomization temperature, 275 °C.

#### 2.2.3. Pharmacokinetic Study of Baicalin in HepG2 Cells

HepG2 cells were cultured at a density of 1×105 cells/mL in 12-well plates for 24 h to facilitate the conduct of cellular pharmacokinetic experiments. The old medium was aspirated and discarded, the wells were rinsed twice with PBS, and 1 mL of 100 μM baicalin and baicalin + PEG400 (0.25%, 0.5%, and 1%) was added into 12-well plates to initiate the timer. The culture was terminated after 0.25 h, 0.5 h, 1 h, 2 h, 4 h, 6 h, 8 h, 10 h, 12 h, and 24 h, respectively. Upon reaching the designated time for administration, the cell culture medium was removed, and the wells were washed with 1 mL of cold PBS solution. Subsequently, 240 μL of ultra-pure water were added to each well, and the sample was stored at −80 °C until analysis. The sample was subjected to sonication in an ice bath for 10 min in order to lyse the cells, thereby enabling the determination of the accumulated drug concentration within the cells. Subsequently, 20 μL of the cell lysate was collected, and the protein concentration was quantified using a BCA kit for normalization.

After repeated freeze-thaw lysis, 100 μL of cell samples were taken, and 300 μL of 0.2% formic acid acetonitrile (containing 200 ng/mL genistein) were added to the vortex for 10 min to precipitate proteins and then centrifuged at 14,000 r/min for 10 min. Then the supernatant was blown dry under nitrogen at 40 °C and then reconstituted with 200 μL of methanol and centrifuged at 14,000 rpm for 10 min. The supernatant was subjected to UPLC-MS/MS analysis.

#### 2.2.4. Western Blot Assay

Western blotting was employed to ascertain the expression of related transporter proteins. HepG2 cells were inoculated in a 6-well plate at a density of 1 × 10^5^ cells per well for 24 h. A series of PEG400 concentrations (prepared with a blank culture medium) were administered for 24 h, after which the cell proteins were collected. Before the western blotting test, the protein concentration was quantified using a bicinchoninic acid (BCA) protein content determination kit. First, the protein was separated by sodium dodecyl sulfate-polyacrylamide gel electrophoresis (SDS-PAGE) on an 8% gel, and then the protein was transferred to a polyvinylidene fluoride (PVDF) membrane. The PVDF membrane was sealed with 5% skimmed milk powder at room temperature for 2 h, after which it was incubated with the first antibody. The main antibodies are as follows: MRP2- antibody, MRP3- antibody, and BCRP- antibody (Shanghai Abmart Biomedical Co., Ltd., Shanghai, China) and Anti-GAPDH (Wuhan servicebio Co., Ltd., Wuhan, China). After incubating with the first antibody at 4 °C overnight, the membrane was washed with TBST buffer, and then incubated with the second antibody at room temperature for 2 h, then washed with TBST. The gel imaging system (Shanghai Tianneng, Shanghai, China) was employed for the detection, utilizing the kernel of heaven and earth and ultra-sensitive ECL luminescence reagent. Finally, the gray value of ImageJ 1.5.2 was employed to assess the expression of the relative protein.

#### 2.2.5. Vesicles Transport Assay

The impact of PEG400 on baicalin transport in vesicles was previously established [30]. Each tube contained 20 μg of vesicles (5 mg/mL), as well as 2.5 mM glutathione (GSH), 100 μM baicalin, PEG400 experimental group (0.25%, 0.5%, and 1%), or a positive control group of 10 μM MK-571 (for MRP2 transporter assay); 2 μM indomethacin (for MRP3 transporter assay); and 6.25 μM Ko143 (for BCRP transporter assay). Prior to the incubation period, the sample was incubated in a solution containing 60 μL of buffer, comprising 250 mM sucrose, 10 mM MgCl_2_, and 10 mM Tris/HCl (pH 7.4), for 15 min at 37 °C. Subsequently, 15 μM Ko143 were added to a centrifuge tube. Subsequently, 15 μL of 25 mM ATP or AMP were added to the centrifuge tubes as test and blank control groups, and then the mixtures were incubated with MRP2 for 5 min, MRP3 for 15 min, and BCRP for 3 min, respectively. After the incubation period, the reaction was terminated by the addition of 200 μL of cryo-wash buffer (40 mM MOPs, 10 mM KCl, pH 7.4). The test solution was then separated from the vesicles using a 0.7 μm glass cellulose membrane, and the filtrate was collected immediately after the membrane was washed three times (200 μL/time). The collected test solution was subsequently analyzed by high-performance liquid chromatography–tandem mass spectrometry (UPLC–MS/MS). The concentration of baicalin in the collected test solution was determined by high-performance liquid chromatography–tandem mass spectrometry (UPLC–MS/MS).

#### 2.2.6. Transport Studies of Baicalin in the MDCK II Cell Model

The cells were inoculated at a density of 0.5 mL (1 × 10^5^) in the upper layer of the transwell plate (AP side), and the culture was changed daily after the lower layer was added with 1.5 mL of complete medium without cells every other day. After 5–7 days of growth, the supernatant was aspirated and discarded, and the cell monolayer was rinsed twice with 37 °C Hank’s Balanced Salt Solution (HBSS, pH 7.4). The electrical resistance of the cell monolayer compartments was measured with a resistivity meter (TEER), and the integrity of the cell monolayer was monitored using TEER. The differentiation of MDCK II-WT, MDCK II-MRP2, MDCK II-MRP3, and MDCK II-BCRP cells was examined by determining sodium fluorescein permeability prior to transit experiments. This was done in order to determine the integrity of the cell monolayer. When the TEER value of the cell monolayer was greater than 150 Ω·cm^2^ and the MDCK II-MRP2, MDCK II-MRP3, and MDCK II-BCRP sodium fluorescein permeability was less than 0.5 × 10^−6^ cm/s or the MDCK II-WT sodium fluorescein permeability was less than 1 × 10^−6^ cm/s, the cell monolayer was considered intact and suitable for transit experiments.
TEER = (TEER_cell monolayer_ − TEER_blank_) × A_area_(1)
wherein TEER_cell monolayer_ is the resistance of the cell monolayer read directly from the Millicell ERS-2 resistivity meter, and TEER_blank_ is the resistance read by inserting a blank HBSS (no cells) only. A_area_ is the surface area of the filter membrane, 1.12 cm^2^.

In this experiment, a 100 μM baicalin solution (dissolved in HBSS) containing different concentrations of PEG400 (0.25%, 0.5%, and 1%) was employed as the test solution, while a solution without PEG400 was used as the blank test solution, which was added to AP or BL chambers, respectively. In the top to basolateral (AP-BL) transport experiment, the 0.5 mL sample added to the apical side (AP) was utilized as the supply room, and the HBSS preheated to 1.5 mL 37 °C was added to the basolateral (BL) as the receiving chamber. Conversely, in the basal low lateral to top (BL-AP) transport experiment, the 1.5 mL sample (dissolved in HBSS) was added to the BL side as the supply room, and the 0.5 mL preheated HBSS was added as a receiving room on the AP side. Once the aforementioned steps had been completed, the transwell was placed on a 37 °C thermostatic oscillator at 60 r/min, and 100 μL was withdrawn from the receiving cell or 10 μL from the supply chamber at 30, 60, 90, and 120 min, depending on the experiment, and replenished with an equal volume of blank 37 °C preheated HBSS. The samples were utilized to precipitate the proteins with 100 μL of methanol containing the internal standard for 10 min. After centrifugation at 16,000× *g* for 15 min at 4 °C, the supernatant was collected in a volume of 70 μL. The contents were determined by UPLC-MS/MS, and apparent permeability coefficients (PAPP) and efflux ratios (ER) were calculated.

The formulas for calculating Papp and ER of monolayer and monolayer are as follows:(2)$$\mathrm{Papp}=\frac{\mathrm{dQ}/\mathrm{dt}}{{\mathrm{AC}}_{0}}$$
wherein dQ/dt (ng/mL·s) is the transit rate (ng/mL·s); A is the membrane area (1.12 cm^2^); and C_0_ (ng/mL) is the initial concentration of the sample in the supply chamber.
ER = Papp_BL-AP_/Papp_AP-BL_
(3)
wherein Papp_BL-AP_ is the apparent permeability coefficient when the BL is added to the sample, and Papp_AP-BL_ is the apparent permeability coefficient when the sample is added at the top side (AP).

#### 2.2.7. Molecular Docking

The crystal structures of docked proteins (BCRP, MRP2, MRP3) were downloaded from the UniProt database. The 3D structure of the small molecule PEG400 was constructed using Chem3D and minimized in the MMFF94 force field. In this study, AutoDock Vina 1.1.2 software was used for molecular docking. Before the commencement of the docking process, PyMol 2.2 was employed to treat the receptor proteins, including the removal of water molecules, salt ions, and small molecules. The docking box was then configured to encompass the entirety of the protein structure. Furthermore, ADFRsuite1.0 was utilized to transform all processed small molecules and receptor proteins into the PDBQT format, which is essential for AutoDockVina1.1.2 docking. When docking, the global search detail is set to 32, and the other parameters are maintained at their default settings. The docking conformation with the highest score is considered the binding conformation. Finally, the PyMol2.5 docking result is employed for visual analysis. 

#### 2.2.8. Statistical Analysis

All data were expressed as the mean ± standard deviation (SD) of triplicate analyses. The DAS 2.0 software, developed by the Mathematical Pharmacology Professional Committee of China, was employed to analyze and statistically process the pharmacokinetic parameters and chronological data of baicalin and its metabolites. Graphs were generated using GraphPad Prism^®^ 9 (GraphPad Software, San Diego, CA, USA), and one-way analysis of variance (ANOVA) was employed to assess the statistical variability of the differences in means. A *p*-value of less than 0.05 was considered to indicate a statistically significant difference, while a *p*-value of less than 0.01 was considered to indicate a highly statistically significant difference.

## 3. Results

### 3.1. Effect of PEG400 on the Cellular Pharmacokinetics of Baicalin

Following the administration of baicalin to cells in the control group and the co-administration of PEG400 with baicalin in the experimental group, the mean blood concentration–time curves were plotted based on the intracellular concentrations of baicalin and metabolites measured at different time points, as illustrated in Figure 1. As illustrated in Table 1, the maximum concentration (C_max_), time to reach maximum concentration (T_max_), and area under the concentration–time curve from time 0 to the last measurable concentration (AUC_0–t_) of baicalin increased in a dose-dependent manner following the administration of baicalin to cells, reaching a maximum absorption concentration of 7.84 μg/mL in combination with PEG400. In comparison to the control group, the area under the concentration–time curve (AUC_0–t_) of baicalin in HepG2 cells exhibited a notable increase from 75.96 μg·h/mL to 106.94 μg·h/mL, 111.97 μg·h/mL, and 130.42 μg·h/mL (*p* > 0.05). Furthermore, the area under the concentration–time curve of baicalin demonstrated a dose-dependent increase, reaching 1.41, 1.48, and 1.72 times the control value. The intracellular baicalein concentration reached a maximum of 0.35 μg/mL at 6 h. The C_max,_ T_max_, and AUC_0–t_ of baicalein after the combined application of low, medium, and high concentrations of PEG400 (0.25%, 0.5%, and 1% PEG400, respectively) increased to varying degrees compared with those of the control group, and the area under the drug–time curve of baicalein in the cells increased from 4.45 μg·h/mL to 5.61 μg·h/mL, 5.48 μg·h/mL, and 5.64 μg·h/mL, respectively, and the AUC_0–t_ of baicalein at low, medium, and high concentrations of PEG400 increased by 1.26-, 1.23-, and 1.27-fold compared with the control group, but not in a dose-dependent manner.

While the intracellular B6G concentration peaked at 0.406 μg/mL at 6 h, the addition of PEG400 resulted in a significant increase in the C_max_, T_max_, and AUC_0–t_ of intracellular B6G, and the area under the drug–time curve increased from 3.62 μg·h/mL to 4.63 μg·h/mL, 4.18 μg·h/mL, and 4.47 μg·h/mL, representing a 1.28-, 1.16-, and 1.24-fold increase over the blank control group, respectively. There was no discernible alteration in the t_1/2_ of baicalin and baicalein, and the discrepancy was not statistically significant (*p* > 0.05). However, the t_1/2_ of B6G was prolonged. These findings indicate that PEG400 enhances the cellular uptake of baicalin in HepG2 cells and facilitates the conversion of baicalin to baicalein with B6G.

### 3.2. Effect of PEG400 on Transporter Protein Expression

In addition, we investigated whether PEG400 could affect the protein expression of transporters involved in baicalin metabolism. Proteins were extracted after treating HepG2 cells with PEG400 at a final concentration of low-medium-high (0.25%, 0.5%, and 1%) (*v*/*v*) for 24 h. The results were summarized as follows. The results of the effect of PEG400 on protein expression are shown in Figure 2.

### 3.3. Vesicle Transport Assay

The transport of baicalin by ABC transporter proteins (MRP2, BCRP, and MRP3) by PEG400 was examined using vesicle transport assays, in which these inside-out vesicles with ATP-binding sites and substrate-binding sites of the transporter face the external buffer and transport the substrate from the outside of the vesicle to the inside. The quantity of substrates that were not transported into the vesicles was quantified by UPLC–MS/MS. The greater the concentration of compounds detected outside the vesicles, the lower the quantity transported, which can be demonstrated by the fact that the treatment can inhibit the corresponding transporters.

In vesicular transport experiments, we investigated the effects of PEG400 on MRP2, MRP3, and BCRP. The efficacy of PEG400 was compared with that of 10 μM MK-571, 6.25 μM Ko143, and 2 μM indomethacin, which were used to observe the degree of activation of MRP3, which may increase the amount of drug entering the bloodstream. Figure 3 illustrates the impact of PEG400 on ATP-dependent baicalin transport in membrane vesicles overexpressing BCRP, MRP2, or MRP3. Overexpression of MRP2, as well as BCRP vesicle groups, showed that PEG400 had different degrees of inhibitory effects at low, medium, and high PEG400 (0.25%, 0.5%, and 1%) doses compared to the blank group, with the highest concentration of baicalin detected outside the vesicles at high concentrations, i.e., stronger inhibition, and higher concentrations of baicalin than the specific inhibitor MK-571 at high concentrations, i.e., PEG400 had better inhibitory effects on MRP2 at both low, medium, and high PEG400 (0.25%, 0.5%, and 1%) doses, and high concentrations all had a better inhibitory effect on MRP2. As illustrated in Figure 3b, the results of the MRP2 vesicle experiment demonstrated that PEG400 exerted a pronounced inhibitory effect on the BCRP transporter, and the high concentration of PEG400 exhibited a more pronounced inhibitory effect than the positive control, exhibiting a superior dose-dependent response at low, medium, and high concentrations. These results collectively demonstrate that PEG400 exerts an inhibitory effect on MRP2 and the BCRP transporter and that PEG400 may cause a decrease in the exocytosis of baicalin and an increase in its intracellular concentration by inhibiting the efflux transporter. However, in contrast, overexpression of MRP3 vesicles demonstrated an inverse trend, whereby the lower the concentration of compounds detected outside the vesicle, the greater the number of compounds transported into the vesicle. Then it can be proved that PEG400 has the activation ability for the corresponding transporter. As illustrated in Figure 3c, the outcomes of MRP3 vesicles demonstrated that PEG400 exhibited varying degrees of activation at low, medium, and high concentrations. Notably, the medium concentration of PEG400 was comparable to that of the positive control. However, it is noteworthy that the activation effect of PEG400 was most pronounced at low concentrations, with the effect becoming progressively weaker at higher concentrations. This phenomenon is indicative of a reverse concentration dependence, whereby the concentration determines the inhibition or activation effect.

### 3.4. Transport Studies in the MDCK II Cell Model

Previous studies have demonstrated the suitability of polarized renal cell monolayers for the study of the transport of ABC transporter proteins that are preferentially localized either apically or basally. To understand whether PEG400 facilitates baicalin uptake by regulating MRP2, BCRP, and MRP3 transport, the present experiments investigated baicalin transport in MDCK II-WT, MDCK II-MRP2, MDCK II-BCRP, and MDCK II-MRP3 cell models.

The effect of PEG400 on baicalin transport was investigated in MDCKII-WT, MDCKII-MRP2, MDCK II-BCRP, and MDCK II-MRP3 cell monolayer models. First, an MTT cell proliferation assay was used to evaluate the effects of baicalin as well as PEG400 concentration on HepG2 cells. As shown in Figure 4, the results indicated that the experimentally selected concentrations of baicalin and PEG400 had no significant effect on the proliferation of the above cells. The resistance value of the cell monolayer and the permeability of sodium fluorescein were in accordance with the requirements, indicating that the integrity of the cell monolayer was good and that it could be used for the transporter experiment. The results of the transporter experiments are shown in Figure 5 and Table 2, which indicate that the exocytosis ratio of MDCKII-WT cells in the control group for the transport of the substrate baicalin was 2.58 ± 0.27. This implies that the transport of baicalin through the cell membrane is mediated by a transporter. For the MDCK-WT cell model, the exocytosis ratios of baicalin decreased to 1.94 ± 0.27, 1.53 ± 0.14, and 1.19 ± 0.12, respectively, in response to low, medium, and high concentrations of PEG400. The apparent permeability coefficients of baicalin from the AP-BL increased in a well-dose-dependent manner, suggesting that PEG400 facilitated the uptake of baicalin into the cell in the basolateral-to-basolateral transporter, i.e., the efflux transporter assay, the apparent permeability coefficient gradually decreased, and the inhibitory effect was strongest at high concentrations. These results suggest that PEG400 inhibits the efflux of baicalin by inhibiting the transporter. Consistent results were obtained in the MDCK II-MRP2 and MDCK II-BCRP cell models, where the percentage of baicalin efflux was reduced, from 2.45 ± 0.04 and 3.01 ± 0.05 to 2.22 ± 0.17, 1.08 ± 0.09, and 0.8 ± 0.04, respectively, and 1.59 ± 0.06, 1.02 ± 0.04, and 0.8 ± 0.02 in a concentration-dependent decrease. The results of the efflux ratio of baicalin in MDCK-WT, MDCK-MRP2, and MDCK-BCRP cell monolayers by 0.25%, 0.5%, and 1% PEG400 are shown in Figure 5f. The efflux ratio of baicalin in MDCK-BCRP cell monolayers was significantly lower when the PEG400 concentration was 0.25% in comparison to MDCK-WT. However, for the MDCK-MRP2 cell monolayer, there was no significant difference in the exocytosis ratio; at 0.5% and 1% PEG400, the efflux ratios of baicalein in both MDCK-MRP2 and MDCK-BCRP cell monolayers were significantly reduced in a concentration-dependent manner, which demonstrated the inhibitory effect of PEG400 on both the transporters MRP2 and BCRP. It is suggested that PEG400 may inhibit the efflux of baicalin by inhibiting the activities of MRP2 and BCRP.

MRP3 is located at the basolateral side and promotes cellular efflux across the basolateral membrane. The increase in the Papp_AP-BL_ and Papp_BL-AP_ ratio of baicalin in MDCK II-MRP3 indicates that MRP3 facilitates the translocation of baicalin to the basolateral side and promotes the entry of the drug into the circulation. In the MDCK II-MRP3 cell model, the PAPP_AP-BL_/PAPP _BL-AP_ in the high-concentration group was 3.76 ± 0.81, which was significantly higher than that in the control group of 0.63 ± 0.37 (Figure 6b), with a 5.97-fold increase in the ratio. It is suggested that PEG400 significantly activated baicalin transport.

### 3.5. Molecular Docking

Molecular docking simulation represents a convenient and effective means of investigating the interaction of small molecules with targets. In this study, we employed the Vina 1.1.2 software to perform docking simulations of PEG400 with baicalin-related transporters, including BCRP, MRP2, and MRP3 proteins. The results of the molecular docking simulations are presented in Table 3. All of the binding energies are negative, indicating that binding is possible for all of the molecules under investigation.

The interaction diagram of the small molecule PEG400 with the efflux transporter protein is shown in Figure 7, from which it can be seen that PEG400 binds to the internal cavity of the protein, respectively. Figure 7a shows the binding pattern between the receptor protein BCRP and the small molecule of PEG400 ligand, where the amino acid residue THR435 forms a hydrogen bond interaction with PEG400. Figure 7b shows the protein interaction pattern between the PEG400 small molecule and the receptor protein MRP2, with amino acid residues SER540, SER1101, ARG1100, and GLN1097 forming hydrogen bond interactions with PEG400. Figure 7c illustrates the binding pattern between PEG400 and MRP3. Amino acid residues SER912, LYS1074, ARG1259, ALA950, and GLY952 form hydrogen bonds with PEG400. Hydrogen bonding is a particularly strong non-covalent interaction, suggesting that the aforementioned amino acids play a significant role in the binding process.

## 4. Discussion

Pharmaceutical excipients are indispensable components of pharmaceutical preparations and usually act as inert substances to influence the disintegration and dissolution of pharmaceutical preparations. In recent years, with the deepening of research, many common pharmaceutical excipients such as surfactants, stabilizers, lubricants, etc. have shown certain physiological and pharmacological activities in vitro and in vivo. A multitude of studies have demonstrated that pharmaceutical excipients can influence the activity of drug transporters and liver enzymes, subsequently affecting the process of drug disposal in vivo. In vitro studies have found that a variety of excipients such as poloxamers [31,32] and polyethylene glycol (PEG) [17,33,34] can inhibit the transport function of efflux transporters.

The majority of flavonoids, such as hyperin and quercetin, are multipolar phenolic hydroxyl compounds like baicalin and exhibit poor hydrophilicity and lipophilicity. In vivo, they were hydrolyzed into corresponding aglycones mainly by intestinal flora and their GUS and then combined with six different subtypes of uridine diphosphate glucuronosyltransferase and sulfate to form glycosides. Baicalein is mainly produced as baicalin under the action of UGT1A9 and B6G under the action of UGT1A8, in addition to the conversion of baicalin by phase II metabolizing enzymes in the liver. Metabolites are also actively transported in and out of the cell by uptake transporters (OATPs) and efflux transporters (BCRPs, MRPs, etc.) in the liver membrane (Figure 8a) [35]. Therefore, the metabolic process of baicalin in the liver is an important link affecting the absorption of baicalin. As a common medicinal excipient, PEG400 is often used in traditional Chinese medicine compound preparations containing baicalin, but whether it affects the pharmacological activity of the drug itself remains to be explored.

Traditional pharmacokinetics is the quantitative study of the macroscopic process of drug absorption, distribution, metabolism, and excretion in organisms. However, the pharmacokinetic behavior of Chinese herbal medicine is different from what we expected in the traditional theory. The determination of the drug concentration in plasma by traditional pharmacokinetics cannot fully explain the concentration and effect of the drug in target organs and target tissues, as well as how the process of drug disposal by cells and the drug concentration at the intracellular target are the decisive factors of drug therapeutic effects [36,37,38]. The traditional approach to pharmacokinetics, which characterizes drug concentration by blood concentration, is inadequate for accurately reflecting the relationship between drug concentration and efficacy at the target site. To address this limitation, pharmacokinetics has been expanded from the traditional level to the cellular level. Furthermore, the cell is considered as a whole in order to quantitatively study the absorption, distribution, metabolism, and excretion of drugs within the cell. This is done in order to elucidate the law of drug disposition within the cell and to predict the targeting and efficacy of drugs within the cell.

The results of the intracellular pharmacokinetic experiments indicated that PEG400 could facilitate the rapid absorption of the substrate drug baicalin, thereby initiating the production of metabolites B and B6G. PEG400 could increase the cellular uptake of baicalin, increase the C_max_, AUC_0–t_, and T_max_ of baicalin and its metabolites B and B6G, decrease the absorption of baicalin in cells, and promote the conversion of baicalin to baicalein, thus producing BG and B6G. It is suggested that PEG400 may activate the related phase II metabolic enzymes. A previous study by the group showed that PEG400 had a significant promoting effect on UGT1A8 and UGT1A9 [39,40].

To further confirm the role of PEG400 in the ADME process of baicalin, this study further employed western blot as well as molecular docking techniques to specifically analyze the effect of PEG400 on the exocytosis transporter associated with the metabolic disposition of baicalin from a molecular perspective. The protein results demonstrated that PEG400 exerted varying degrees of up- or downregulation of the efflux transporter, and baicalin influenced the protein expression of the transporter in HepG2 cells, which was consistent with the findings of previous experiments. The molecular docking results indicated that PEG400 all had binding sites within the protein cavity when interacting with the transporter, suggesting that PEG400 may occupy the binding sites within the protein, thus acting as an inhibitor at a macroscopic level.

Transporters and metabolizing enzymes are involved in the intestinal and hepatic absorption and metabolism of numerous clinical drugs and may affect the pharmacokinetics of co-administered drugs [41,42]. To explore the possible mechanisms of the effects of PEG400 on baicalin cell pharmacokinetics and the reasons for its low bioavailability, the MDCKII-BCRP, MDCKII-MRP2, and MDCKII-MRP3 cell monolayer models (Figure 8c), as well as membrane vesicles overexpressing specific transporter proteins (BCRP, MRP2, and MRP3) (Figure 8b), in combination with exocytosis of transporter-specific inhibitors, were used to investigate the effects of PEG400 on the transporters. Furthermore, western blot, molecular docking, and other techniques were employed to investigate the effects of protein expression of the major transporters of baicalin (BCRP, MRP2, and MRP3) and their potential mechanisms of action [43].

Changes in the expression and function of transporters frequently result in alterations to pharmacokinetics, which represents the primary focus of drug interactions [15,44]. Baicalin has been identified as a substrate for the efflux transporters MRP2, MRP3, and BCRP; thus, we proceeded to investigate the direct transport of baicalin through MRP2, MRP3, and BCRP vesicles and to ascertain whether PEG400 affects the relevant efflux transporters and, as a consequence, affects the metabolic kinetics of baicalin in HepG2 cells.

Furthermore, the inhibitory effect of PEG400 on transporter-mediated transport in comparison to specific inhibitors was evaluated. The results demonstrate that PEG400 inhibits baicalin efflux or translocation to the interior of vesicles in a concentration-dependent manner in BCRP and MRP2 vesicular transport in vitro, with a stronger inhibitory effect than the specific inhibitors at high concentrations. However, the results differed in MRP3 vesicles, inhibiting at low concentrations and promoting the transport of baicalin into the vesicles. Interestingly, although baicalin was inhibited compared with the control group at middle and high doses, the baicalin detected outside the vesicles showed an upward trend, which may be the two-way effect of PEG400, that is, the concentration of PEG400 determines the final inhibition or activation. This indicates that it is crucial to consider the concentration range when utilizing PEG400 with MRP3.

Previous studies have demonstrated the suitability of polarized renal cell monolayers for the study of ABC transporter transport, which is preferentially located at the top or base [45,46]. In the present study, an in vitro cell culture model of an MDCK II cell line stably and highly expressing human MRP2, MRP3, and BCRP was established by introducing human MRP2, MRP3, and BCRP genes into MDCK II cells [47,48]. To understand whether PEG400 promotes the uptake of baicalin from lamps by regulating the MRP2, BCRP, and MRP3 transporters, this experiment investigated baicalin transport in the MDCK II-WT, MDCK II-MRP2, MDCK II-BCRP, and MDCK II-MRP3 cell models. From the results of baicalin in the cell monolayer model, it is known that there are transporters involved in mediating the transport process of baicalin and that baicalin is a substrate for the transporters MRP2, MRP3, and BCPR. Vesicular transport was consistent with the results of transmembrane transport experiments in cell monolayers, both of which acted in a concentration-dependent manner at low to medium-high concentrations of PEG400. 

However, in the MDCK II-MRP3 cell model, which differed from the results of vesicular transport, the greatest amount of baicalin was secreted to the basolateral side at high concentrations. This is probably due to the cell monolayer model; PEG400 selectively increased the permeability of the basolateral cell membrane, thereby increasing the transport of baicalin to the basolateral side. The results of both studies indicate that PEG400 may facilitate baicalin uptake by simultaneously activating MRP3 and inhibiting MRP2 and BCRP, thereby increasing the maximum intracellular uptake of baicalin as well as the area under the pharmacophore curve and thus increasing the bioavailability of baicalin.

Possible theories for PEG400 affecting baicalin bioavailability include PEG400 altering cell membrane integrity or fluidity, competitively blocking binding sites, interfering with ATP hydrolysis, or altering the expression of efflux transporter proteins [49,50,51]. Some pharmaceutical excipients have been reported to reduce the lipid mobility of cell membranes and fluidize the cellular lipid bilayer, which may result in changes to the secondary and/or tertiary structure of membrane proteins. This, in turn, can lead to loosening of the phospholipid bilayer and alterations in its biological activity [52]. Alternatively, it can penetrate the plasma membrane and inhibit transporter proteins through a membrane fluidization mechanism [53,54]. It is well known that PEG400, as a solvent, solubilizer, O/W emulsifier, and stabilizer, is commonly used in soft gels, injectables, solid lipid nanoparticles, etc. PEG400 can form intramolecular and intermolecular H-bonds through its terminal hydroxyl group, which has high polarity, increasing hydrophilic and hydrophobic interactions with hydrophobic drugs, which is an important property for solubilization [55,56]. Therefore, as a solubilizer, PEG400 can increase the solubility of baicalin, which may also be one of the reasons for the increase in bioavailability of baicalin, which may make it possible for baicalin to accumulate in cells. These potential mechanisms provide a theoretical basis for the hypothesis that PEG400 may have the potential to promote oral absorption in vivo.

## 5. Conclusions

In conclusion, the addition of the pharmaceutical excipient PEG400 will make the metabolic disposition process of baicalin in the cell change, confirming the interaction between PEG400 and the efflux transporter, and the study lays the research foundation for the pharmaceutical excipient–drug interaction, based on which it also provides a new paradigm for the scientific use of the excipient PEG400. Most importantly, the fact that different dosages of PEG400 in pharmaceutical formulations produce different effects deserves further consideration.

## Figures and Tables

**Figure 1 pharmaceutics-16-00731-f001:**
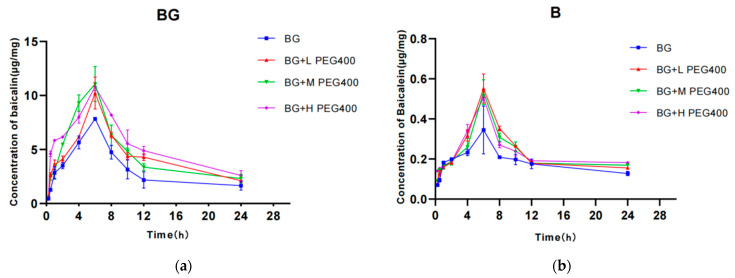
Effect of PEG400 on the pharmacokinetics of baicalin in hepatocytes. Concentration-time profiles of baicalin (**a**), baicalein (**b**), and B6G (**c**). (**d**) AUC of area under the drug–time curve for BG. (**e**) AUC of area under the drug–time curve for B and B6G. Compared to the control group, ** *p* < 0.01; *** *p* < 0.001. (mean ± SD, *n* = 3).

**Figure 2 pharmaceutics-16-00731-f002:**
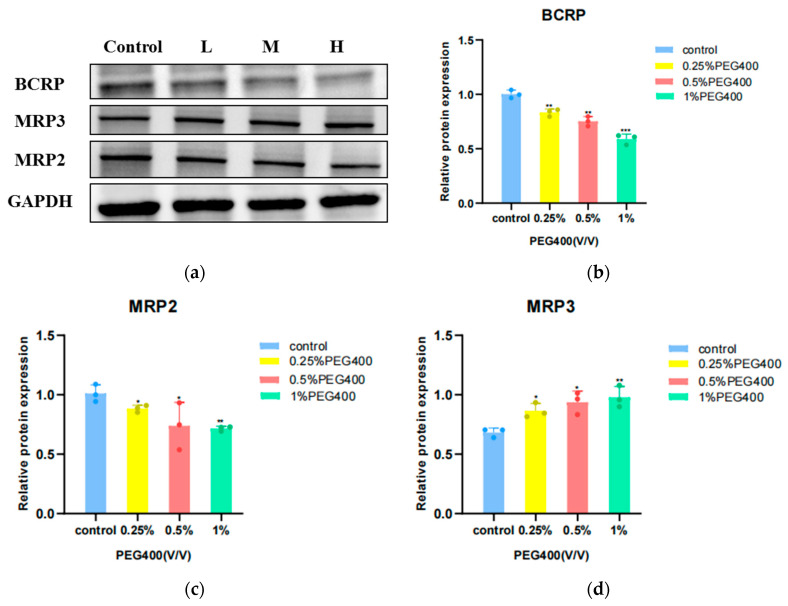
Effect of PEG400 on representatives of baicalin-related transporters. (**a**) Protein bands of MRP2, BCRP and MRP3 transporters; (**b**) the protein expression of MRP2; (**c**) the protein expression of BCRP; (**d**) the protein expression of MRP3. Compared to the control group, * *p* < 0.05; ** *p* < 0.01; ****p* < 0.001. (mean ± SD, *n* = 3).

**Figure 3 pharmaceutics-16-00731-f003:**
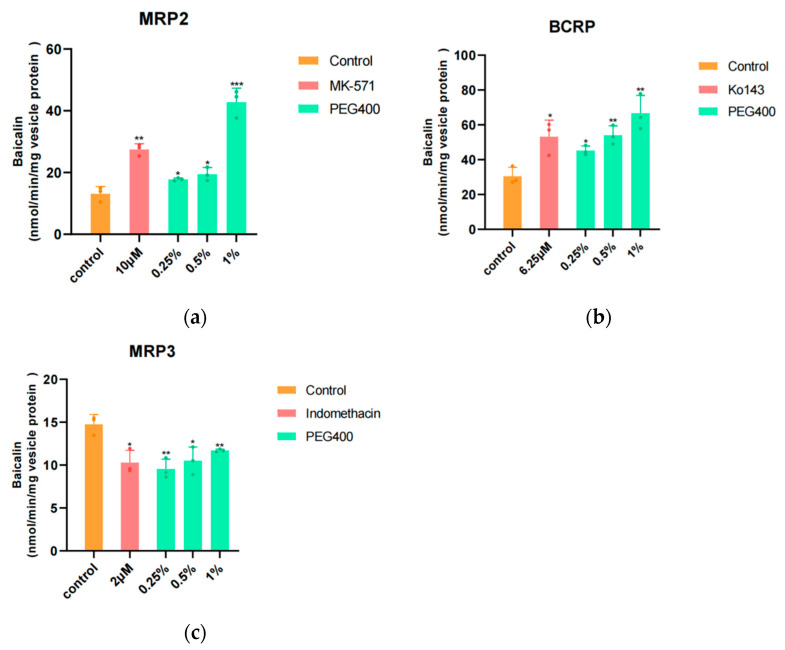
Effect of PEG400 on baicalin vesicular transport in Sf9 insect cells with high expression of BCRP, MRP2, or MRP3. (**a**) Effects of MRP2 inhibitors MK-571 and PEG400 on baicalin vesicular transport in Sf9 insect cells with high expression of MRP2. (**b**) Effects of BCRP inhibitors Ko143 and PEG400 on baicalin vesicular transport in Sf9 insect cells overexpressing BCRP. (**c**) Effects of MRP3 activator indomethacin and PEG400 on baicalin vesicle trafficking in Sf9 insect cells with high expression of MRP3. Compared to the control group, * *p* < 0.05; ** *p* < 0.01; *** *p* < 0.001. (mean ± SD, *n* = 3).

**Figure 4 pharmaceutics-16-00731-f004:**
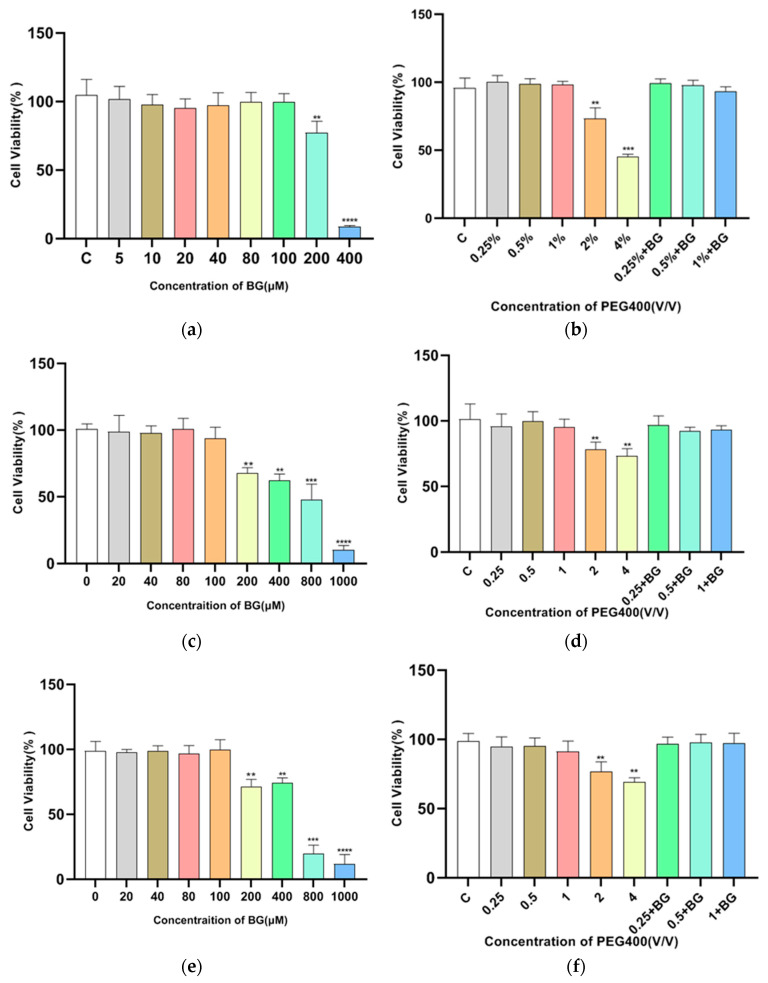
The effect diagram of cell vitality. (**a**,**b**) The effects of different concentrations of baicalin and PEG400 on the survival of MDCK II-WT cells. (**c**,**d**) The effects of different concentrations of baicalin and PEG400 on the survival of MDCK II-MRP2 cells. (**e**,**f**) The effects of different concentrations of baicalin and PEG400 on the survival of MDCK II-BCRP cells. (**g**,**h**) The effects of different concentrations of baicalin and PEG400 on the survival of MDCK II-MRP3 cells. Compared to the control group, ** *p* < 0.01; *** *p* < 0.001, **** *p* < 0.0001. (mean ± SD, *n* = 5).

**Figure 5 pharmaceutics-16-00731-f005:**
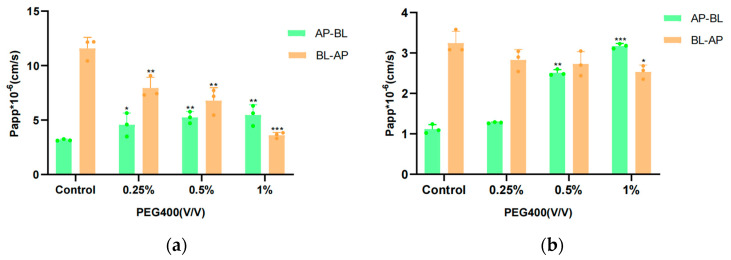
Effect of PEG400 on baicalin transport across the epithelium via cell monolayers. Apparent permeability coefficient Papp values for baicalin transport in the MDCK II-WT (**a**); MDCK II-MRP2 (**b**); MDCK II-BCRP (**c**); and MDCK II-MRP3 (**d**) cell monolayer models. Efflux ratios of baicalin in cell monolayer models (**e**); Comparison of the efflux ratio of PEG400 to baicalin in overexpressing MRP2 and BCRP cells with MDCK-WT (**f**). (mean ± SD, *n* = 3). * *p* < 0.05; ** *p* < 0.01; *** *p* < 0.001, ns *p* > 0.05.

**Figure 6 pharmaceutics-16-00731-f006:**
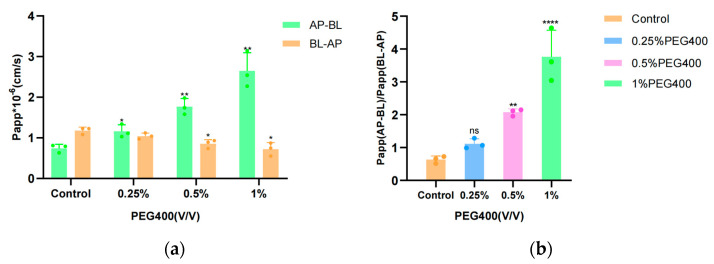
The effect of PEG400 on transepithelial transport of baicalin from AP to BL side and BL to AP side across MDCK II-MRP3 cell monolayer. (**a**) Values of Papp_AP-BL_ and Papp_BL-AP_ for baicalin in the MDCK II-MRP3 cell model. (**b**) The ratio of PAPPAP-BL to PAPPBL-AP for baicalin in the MDCK II-MRP3 cell model. Compared to the control group, * *p* < 0.05; ** *p* < 0.01; **** *p* < 0.0001; ns *p* > 0.05. (mean ± SD, *n* = 3).

**Figure 7 pharmaceutics-16-00731-f007:**
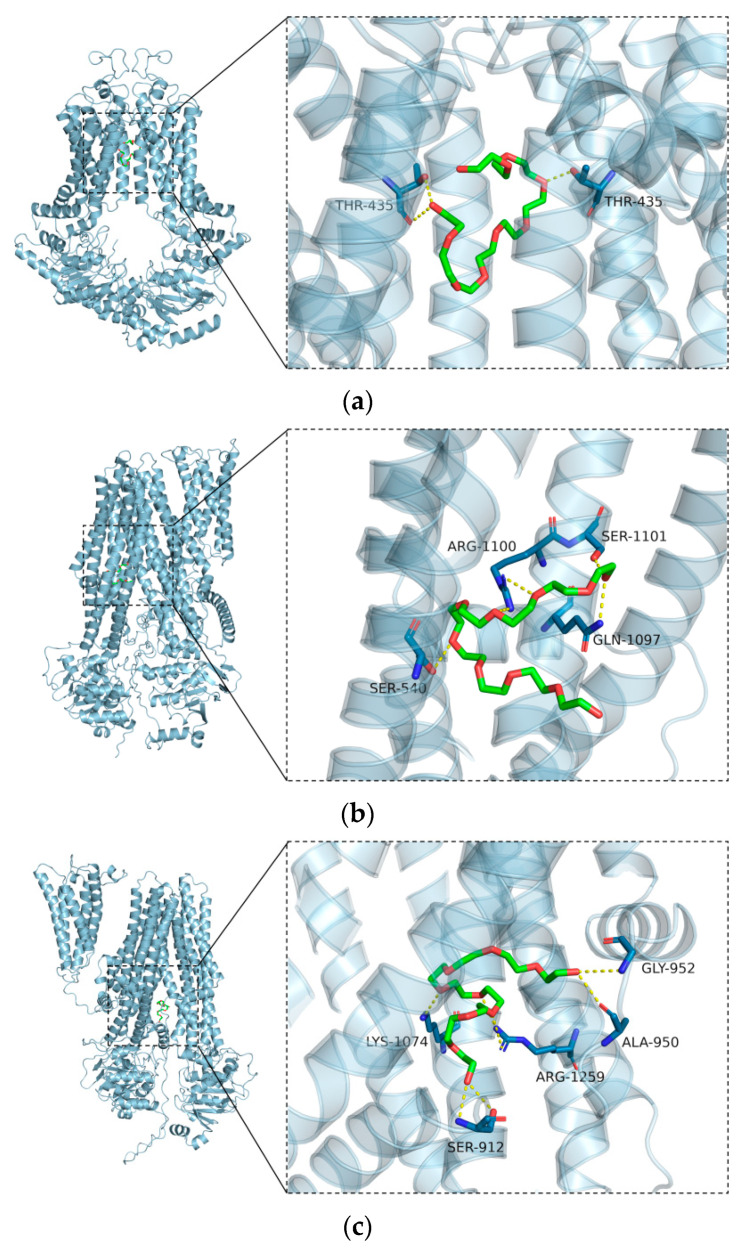
The binding pattern of the small molecule PEG400 with the efflux transporter proteins BCRP (**a**), MRP2 (**b**), and MRP3 (**c**) based on the docking obtained from the docking, with an overall view on the left and a localized view on the right, where the amber stick is the small molecule, the cyan cartoon is the protein, the red-green structure is the PEG400 molecule, and the yellow dashed line indicates the hydrogen-bonding interaction.

**Figure 8 pharmaceutics-16-00731-f008:**
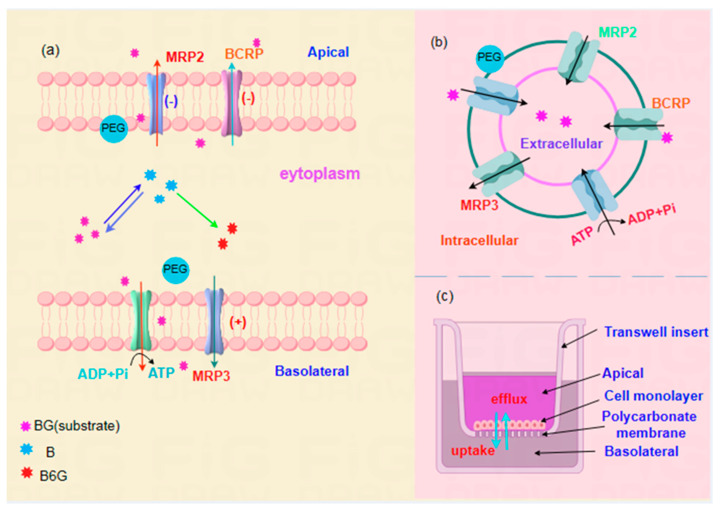
Schematic representation of possible molecular mechanisms by which efflux transporter-based PEG400 affects the metabolic disposition of baicalin in cells (**a**). Schematic diagram of experimental mechanism of vesicle transport (**b**). Schematic diagram of MDCK cell model transwell chambers (**c**).

**Table 1 pharmaceutics-16-00731-t001:** Pharmacokinetic parameters of baicalin and its metabolites in cells (mean ± SD, *n* = 3).

Parameters	Groups	BG	B	B6G
AUC_0–t_ (μg·h/mL)	BG	75.96 ± 2.57	4.45 ± 0.06	3.62 ± 0.27
	BG + L PEG400	106.94 ± 2.22	5.61 ± 0.09	4.63 ± 0.04
	BG + M PEG400	111.97 ± 3.987	5.48 ± 0.118	4.18 ± 0.13
	BG + H PEG400	130.42 ± 5.26	5.64 ± 0.07	4.47 ± 0.14
t_1/2_ (h)	BG	6.77 ± 5.92	17.63 ± 7.99	5.19 ± 2.37
	BG + L PEG400	9.91 ± 3.76	3.73 ± 0.54	6.08 ± 1.68
	BG + M PEG400	6.02 ± 4.23	9.40 ± 8.73	13.97 ± 2.34
	BG + H PEG400	5.43 ± 0.91	6.43 ± 1.85	8.55 ± 1.18
T_max_ (h)	BG	6.00	6.00	6.00
	BG + L PEG400	6.00	6.00	6.00
	BG + M PEG400	6.00	6.00	6.00
	BG + H PEG400	6.00	6.00	8.00
C_max_ (μg/mL)	BG	7.84 ± 0.11	0.35 ± 0.12	0.41 ± 0.04
	BG + L PEG400	10.23 ± 1.48	0.55 ± 0.07	0.51 ± 0.08
	BG + M PEG400	11.17 ± 1.50	0.53 ± 0.06	0.58 ± 0.06
	BG + H PEG400	10.81 ± 0.35	0.51 ± 0.01	0.56 ± 0.03

**Table 2 pharmaceutics-16-00731-t002:** The apparent permeability coefficient of BG on MDCK II-WT, MDCK II-BCRP, MDCK II-MRP2, and MDCK II-MRP3 monolayers (mean ± SD, *n* = 3).

Cell Models	Groups	Papp (×10^−6^ cm/s)		ER
		AP-BL	BL-AP	
MDCK II-WT	BG	3.18 ± 0.07	11.58 ± 1	3.64 ± 0.28
	BG + 0.25%PEG400	4.57 ± 1.08	7.94 ± 0.99	1.8 ± 0.44
	BG + 0.5%PEG400	5.24 ± 0.53	6.78 ± 1.19	1.29 ± 0.11
	BG + 1%PEG400	5.47 ± 0.94	3.61 ± 0.26	0.67 ± 0.07
MDCK II-BCRP	BG	3.43 ± 0.07	10.34 ± 0.02	3.01 ± 0.05
	BG + 0.25%PEG400	5.64 ± 0.48	8.98 ± 0.42	1.59 ± 0.06
	BG + 0.5%PEG400	7.4 ± 0.43	7.51 ± 0.47	1.02 ± 0.04
	BG + 1%PEG400	8.59 ± 0.19	6.9 ± 0.04	0.8 ± 0.02
MDCK II-MRP2	BG	1.12 ± 0.11	3.28 ± 0.27	2.45 ± 0.04
	BG + 0.25%PEG400	1.27 ± 0.02	2.83 ± 0.26	2.22 ± 0.17
	BG + 0.5%PEG400	2.51 ± 0.07	2.73 ± 0.31	1.08 ± 0.09
	BG + 1%PEG400	3.17 ± 0.06	2.53 ± 0.17	0.8 ± 0.04
MDCK II-MRP3	BG	0.74 ± 0.1	1.18 ± 0.08	0.63 ± 0.37
	BG + 0.25%PEG400	1.06 ± 0.16	1.04 ± 0.08	1.11 ± 0.15
	BG + 0.5%PEG400	1.77 ± 0.2	0.85 ± 0.1	2.08 ± 0.1
	BG + 1%PEG400	2.65 ± 0.44	0.72 ± 0.16	3.76 ± 0.81

**Table 3 pharmaceutics-16-00731-t003:** Docking results of PEG400 with efflux transporter proteins.

Ligand_Name	Target_Name	Binding Energy (kcal/mol)
PEG400	BCRP	−4.8
PEG400	MRP2	−4.4
PEG400	MRP3	−4.2

## Data Availability

The data presented in this study are available in this article and Appendix A.

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
