# Peer review of "Study on the Effect of Pharmaceutical Excipient PEG400 on the Pharmacokinetics of Baicalin in Cells Based on MRP2, MRP3, and BCRP Efflux Transporters"

_pharmaceutics, 2024, doi:10.3390/pharmaceutics16060731_

Round 1
Reviewer 1 Report (Previous Reviewer 2)
Comments and Suggestions for Authors
no comments
Comments on the Quality of English LanguageNeed to be proof read by an English writer to check for grammar.
Author Response
Dear Reviewer,
I sincerely appreciate your diligent review of our submitted manuscript and the valuable feedback and suggestions provided. Your professional insights have greatly assisted our research and prompted us to delve deeper into refining our work.
Comments and Suggestions for Authors
no comments
Comments on the Quality of English Language
Need to be proof read by an English writer to check for grammar.
Response: Dear reviewer, thank you for your feedback. We have taken seriously the question about your pointing out that the English language needs to be proofread by an English writer to check grammar, and we have made corrections with the help of a proofreader who is a native English speaker, checking grammar. Your insights are invaluable, and we are committed to improving the overall quality of the manuscript.

Reviewer 2 Report (New Reviewer)
Comments and Suggestions for Authors
The present manuscript (Pharmaceutics-3001409), entitled "Study on the effect of pharmaceutical excipient PEG400 on the pharmacokinetics of baicalin in cells based on MRP2, MRP3 and BCRP efflux transporters", by Dan Yang et al, refers to the authors' studies on the effect of PEG400 on the metabolic kinetics of baicalin (5,6-dihydroxy-4-oxoflav-2-en-7-yl β-D-glucopyranosiduronic acid, BG) at the hepatocyte level, using human-derived hepatocellular carcinoma (HepG2) cells. The changes in the content of baicalin and its metabolites were determined in HepG2 cells, by utilizing Western blot and other techniques to investigate the effect of protein expression of the major transporters of baicalin (BCRP, MRP2, and MRP3) and thus its possible mechanism of action. The effects of PEG400 on MRP2, MRP3 and BCRP efflux transporters were investigated by the vesicular transport experiments of MRP2, MRP3 and BCRP and the cell models of MDCKII- WT, MDCKII-MRP2, MDCKII-MRP3 and MDCKII-BCRP.
The requisite methodologies/assays were followed to decipher whether the addition of the excipient PEG400 in different dosages can increase the solubility of baicalin, and be one of the reasons for the increase of baicalin's bioavailability. The authors conclude that their findings provide a theoretical basis for the hypothesis that PEG400 may have the potential to promote oral absorption in vivo.
Although, as already mentioned, the authors utilized the appropriate methods/assays, en route to the implementation of their study, the discussion on the enhancement of baicalin's solubility, due to PEG400, is limited.
It is well known that PEG400 forms both intra- and intermolecular H-bonds via its end hydroxyl groups. Owing to this property, PEG400 promotes the solubility of molecules bearing H-bond donor and H-bond acceptor functionalities, like the C5, C6 OHs of baicalin and the 3, 4, 5 OHs of its aglycone baicalein portion.
Thus, in the revised version of the article, the authors have to take these points into account and enrich the Discussion section, accordingly, by adding relevant References. Moreover, the revised version has to be proof-read by a native English speaking person, as there are considerable English language flaws.
Comments on the Quality of English LanguageThe revised version has to be proof-read by a native English speaking person, as there are considerable English language flaws.
Author Response
Dear Reviewer,
We sincerely appreciate your diligent review of our submitted manuscript (Pharmaceutics-3001409) and the valuable feedback and suggestions provided. Your professional insights have greatly assisted our research and prompted us to delve deeper into refining our work.
Comments 1:The present manuscript (Pharmaceutics-3001409), entitled "Study on the effect of pharmaceutical excipient PEG400 on the pharmacokinetics of baicalin in cells based on MRP2, MRP3 and BCRP efflux transporters", by Dan Yang et al, refers to the authors' studies on the effect of PEG400 on the metabolic kinetics of baicalin (5,6-dihydroxy-4-oxoflav-2-en-7-yl β-D- glucopyranosiduronic acid, BG) at the hepatocyte level, using human-derived hepatocellular carcinoma (HepG2) cells. The changes in the content of baicalin and its metabolites were determined in HepG2 cells, by utilizing Western blot and other techniques to investigate the effect of protein expression of the major transporters of baicalin (BCRP, MRP2, and MRP3) and thus its possible mechanism of action. The effects of PEG400 on MRP2, MRP3 and BCRP efflux transporters were investigated by the vesicular transport experiments of MRP2, MRP3 and BCRP and the cell models of MDCKII- WT, MDCKII-MRP2, MDCKII-MRP3 and MDCKII-BCRP.
The requisite methodologies/assays were followed to decipher whether the addition of the excipient PEG400 in different dosages can increase the solubility of baicalin, and be one of the reasons for the increase of baicalin's bioavailability. The authors conclude that their findings provide a theoretical basis for the hypothesis that PEG400 may have the potential to promote oral absorption in vivo.
Although, as already mentioned, the authors utilized the appropriate methods/assays, en route to the implementation of their study, the discussion on the enhancement of baicalin's solubility, due to PEG400, is limited.
It is well known that PEG400 forms both intra- and intermolecular H-bonds via its end hydroxyl groups. Owing to this property, PEG400 promotes the solubility of molecules bearing H-bond donor and H-bond acceptor functionalities, like the C5, C6 OHs of baicalin and the 3, 4, 5 OHs of its aglycone baicalein portion.
Thus, in the revised version of the article, the authors have to take these points into account and enrich the Discussion section, accordingly, by adding relevant References. Moreover, the revised version has to be proof-read by a native English speaking person, as there are considerable English language flaws.
Response: Dear reviewer, Thank you for your valuable comments. Regarding your suggestion to add a note related to increasing solubility, we understand your point of view and recognize that it may not have been adequately expressed in the manuscript. I have added the points to the Discussion section and highlighted them in blue font, as well as added the relevant references. The added discussion section is below:
"It is well known that PEG400, as a solvent, solubilizer, O/W emulsifier, and stabilizer, is commonly used in soft gels, injectables, solid lipid nanoparticles, etc. PEG400 can form intramolecular and intermolecular H-bonds through its terminal hydroxyl group, which has high polarity increasing hydrophilic and hydrophobic interactions with hydrophobic drugs, which is an important property for solubilization [58, 59]."
In addition, we take seriously the questions about the flaws in the English language that you have pointed out, and thoroughly proofread with the help of native English speakers to effectively address these issues to improve readability. Your insights are invaluable, and we are committed to improving the overall quality of the manuscript.

Reviewer 3 Report (New Reviewer)
Comments and Suggestions for Authors
I have gone through the entire manuscript. The writing part was very impressive. The authors have written a good introduction. Maintained very good flow in the entire research article. Impact of PEG 400 with baicalin exploration studies were explored in this manuscript. Methods and protocols were designed appropriately. In my opinion, the submitted research article should go through minor revision. The comments are mentioned below, please address them accordingly.
In 2.2.2à Authors kindly include chromatograms images.
In 2.2.2à Please specify what make and model UPLC and Mass specs were used.
In 2.2.2à XBridge BEH C18 column (2.1x100 mm)- 18 should be subscript.
In 2.2.3à Authors do you have any explanation why HepG2 cell lines were used instead of animals for evaluating pharmacokinetic parameters? Also, how, or what software was used for calculating pharmacokinetic parameters. Kindly include the software details.
In 3.1à Please define what are low, medium, and high concentrations?
In table:1à I noticed there was a pattern with AUC and Cmax. with higher the PEG concentrations, higher the AUC and Cmax. So, the query is why authors have not chosen PEG concentrations beyond 1% any logical explanation for this?
Author Response
Dear Reviewer,
We sincerely appreciate your valuable time and effort in carefully reviewing our submission and providing us an opportunity to improve the quality of our submitted manuscript (Pharmaceutics-3001409). In this revision, we have addressed all the comments/suggestions made and highlighted them in blue in the text, and we hope that the revised manuscript is now in line with the publication standards of your journal.
Comments 1: In 2.2.2à Authors kindly include chromatograms images.
Response 1: The original chromatograms of baicalin, its metabolites and the internal standard (IS: genistein) have been added to the supplementary material.
Comments 2: In 2.2.2à Please specify what make and model UPLC and Mass specs were used.
Response 2: The brand and model information of UPLC and MASS has been added to the text as shown in the attached chart.
Comments 3: In 2.2.2à XBridge BEH C18 column (2.1x100 mm)- 18 should be subscript
Response 3: Thank you for your feedback and guidance. I have made a revision in the manuscript, changing “18” to subscript format.
Comments 4:In 2.2.3à Authors do you have any explanation why HepG2 cell lines were used instead of animals for evaluating pharmacokinetic parameters? Also, how, or what software was used for calculating pharmacokinetic parameters. Kindly include the software details.
Response 4: Dear reviewers, the preliminary research group on pharmacokinetics of baicalin in rat plasma has been published. Pharmacokinetics of drugs in plasma can not fully explain the pharmacological effects of drugs, and there are some limitations in accurately predicting the efficacy of drugs in vivo. Generally speaking, the effect of drugs is that they reach the cell through multiple biological barriers, interact with many specific targets in the cell, and produce related biological effects. The drug concentration around the target in the cell and the plasma drug concentration can more accurately reflect the drug effect. Hepatocytes contain most enzyme systems, including phase I and phase II heterogeneous metabolic systems, and various transporters. Therefore, human hepatocytes are considered as the gold standard for predicting human pharmacokinetics in drug metabolism research. In our study, in order to investigate whether the pharmacokinetic effects of PEG400 on baicalin in cells were consistent with the pharmacokinetic results in rat plasma, with a view to fully elucidating the effects of PEG400 on the bioavailability of baicalin. In addition, the pharmacokinetic parameters were analyzed by DAS 2.0 (Mathematical Pharmacology Professional Committee of China) data statistics software, and the software information was added to the manual modification.
Comments 5: Please define what are low, medium, and high concentrations?
Response 5: In 3.1, the low, medium and high concentrations of PEG400 are 0.25%, 0.5% and 1%(V/V). Thank you for your valuable comments, and I have added explanations in the manuscript.
Comments 6: In table:1à I noticed there was a pattern with AUC and Cmax. with higher the PEG concentrations, higher the AUC and Cmax. So, the query is why authors have not chosen PEG concentrations beyond 1% any logical explanation for this?
Response 6: Dear reviewer, Thank you for your valuable feedback. In this manuscript, the pharmacokinetic parameters of AUC and Cmax increased with increasing concentration of PEG400, but according to the results of MTT experiments done in the pre-experimental period, it was shown that the inhibition of HepG2 cells started when the concentration of PEG400 was 2%. Therefore,we have not chosen PEG400 concentrations beyond 1%, and the results of the MTT experiments are shown in Figure 2 of the Supplementary Material.

Round 2
Reviewer 2 Report (New Reviewer)
Comments and Suggestions for Authors
In the revised version of the pharmaceutics-3001409 manuscript, the authors have successfully dealt with the points raised in my initial review.
Author Response
Dear Reviewer,
We sincerely appreciate your valuable time and effort in carefully reviewing our submission and providing us an opportunity to improve the quality of our submitted manuscript (Pharmaceutics-3001409).
Comments: In the revised version of the pharmaceutics-3001409 manuscript, the authors have successfully dealt with the points raised in my initial review.
Response: Dear reviewers, Thank you very much for your valuable comments and feedback on our manuscript, as well as for recognizing our manuscript revision efforts. Wish you all the best!

This manuscript is a resubmission of an earlier submission. The following is a list of the peer review reports and author responses from that submission.
Round 1
Reviewer 1 Report
Comments and Suggestions for Authors
Dear authors,
with interest I read your publication draft. There are some small comments (see below) but also a general aspect/question from my site which I would like to start my review report.
Your aim was to add detailed information about the influence of PEG400 as an excipient esp. for the uptake/efflux of bacalein on the cellular level.
1) You published something comparable on a different cell model. Why does it make sense to use the ones here? What makes it better/improved?
2) You seem to be a little bit sceptical about "classical" pharmakokinetics, at least regarding your API of interest. Therefore, I was wondering how does your results fit in a more global view? You are right, that cellular uptake plays an important role but also biodistribution via our central compartiment (blood) is necessary and thus plasma/blood concentration are important. In other words: For which in vivo aspect are your cell models relevant? Absorption in the GIT into the blood followed by biodistribution/elimination? Or more like the target cells and how they handle the API? Is the PEG400 at the same place at the same time? etc.
lines 46/47: something missing here? sentence does not make sense...
lines 83-86: in vitro vs. intestinal cavity/systemic circulation - does this fit?
line 126: which quality does the PEG400 had?
lines 138+: did you check/identify the cell line type regularily? esp. an concern when cells are given from lab to lab...
lines 162+: do you really lyse the cells in the tissue culture wells? or how do you move them quantitatively from the wells into the freezing tubes?
lines 180+: which antibodies did you use?
line 229: is the equation correct? brackets missing?
line 237: bracket missing!
lines 272+: ttests are used to compare two groups of nromal distributed data --> are your data normal distributed? you have more then 2 groups, so, is ttest really good, menaing powerful, for your analysis?!?
lines 284-285: which difference? which parameter did you compare? with a simple ttest that does not work out...
lines 309+: Fig 1 - d is missing?! to small for a good reading; hwy did you use these time points? risk of missing the real t_max/c_max and uncertainty in t1/2...
lines 336+: did you checked literature for information about PEG400 and protein expression (in other cell lines, from other labs/groups...)? one could expect something since PEG400 is used quite often/over years...
line 402: of of
lines 406/407: data for cell viability? maybe as supplement?
line 441: target targets?!?
discussion part: how do you judge your results in vitro in context of exspected in vivo results? invitro/invivo correlation of your observation - could you comment on that? maybe a bit of speculation...
line 541: Fig7(a) - did I get it right - ATP is extra cellulary produced at the basolateral site by the efflux of your API?
line 546: 4?
line 592: PEG400 has solubilization and in creases... something missing here, or?
lines 594-596: why does this observation of an cellular accumulation leads to a better absorption in GIT?
Kind regards.
Reviewer 2 Report
Comments and Suggestions for Authors
Comments:
In the introduction, the author did not clearly provide any literature or rationales explaining why PEG400 would potentially affect drug transporter protein expression nor that PEG400 concentrations studied had any clinical relevance. If taken orally, no evidence that PEG400 levels would be readily absorbed (bioavailability was unknown and not stated). In addition, no scientific rationale that PEG400 would interact with transcription factors that will alter protein expression.
For Section 3.1, the experiment conducted with HepG2 was scientifically flawed. In principle, the use of these HepG2 cells is used for the study of cellular efflux or accumulation and not a pharmacokinetic study. The cells would be incubated with the presence of study substrate for fixed amount of time where the study substrate concentration is at steady state (both in media and in the cell), then rinsed/washed and incubated with PEG400 to measure whether the efflux of the study substrate is inhibited or not. In addition, expressing PEG400 as a percentage instead of PEG400 concentration had no clinically meaningfulness to the study objective. Based on the way the study was conducted, the author had not provided data that PEG400 did not bind with BG directly to elute the results.
For Section 3.2, the experiment is flawed. No rationale why the PEG400 is accumulated for 24 hours as it takes at least a few days to see any protein changes of any drugs. In addition, why are mRNA levels not measured?
For Section 3.3, rationales would need to be provided in the discussion as protein expression of BCRP and MRP2 in Figure 3 are in the opposite direct as compared to the results in Figure 2.
For Section 3.4, the results of the efflux study (Figure 4) did not provide any evidence that PEG400 had any inhibition on MRP2 nor BCRP. The efflux ratio results should be compared between 0.25% PEG400 in MDCK-WT vs. 0.25% PEG400 in MDCK-MRP2 vs. 0.25% PEG400 in MDCK-BCRP. Same analyses would be performed with 0.5% PEG400 and 1% PEG400. No apparent difference between MDCK-WT, -MRP2, and -BCRP were observed with the same concentration of PEG400. Overexpression of MRP2 and BCRP in MDCK would clearly show a difference in ER if there is some type of drug transporter inhibition by PEG400. However, the results did not support the findings.
Comments on the Quality of English Language
Scientific writing needs to be further improved. eg. "More and more studies" is not proper English.
When you are saying that a drug has poor bioavability or poorly absorbed, it is not the same as malabsorption, which is what you say on line 62-63.
Reviewer 3 Report
Comments and Suggestions for Authors
In the methods section, please provide the details of the mobile phase used for LCMS method.
Please insert one representative chromatogram of LCMS including the target drug and its metabolites.
The centrifuge speed should be presented in standard format. (either rpm or g).
The equation should be given numbers.
The p-values should also be incorporated in the tables.
The p-values described in the legend of figure 1 are not potrayed in the figure. please correct the legend description.
In table 1, tmax is not presented as mean +/- SD.
In table, the half-life BG+M PEG400, for B, the value is 9.40±8.73, the standard deviation is slightly high. It is acceptable but the author can recheck and verify it.
The y-axis of figure 1 are labelled as µg/mg. It is probably µg/ml. Please correct it.
The author has used x±s and mean ± SD format. Please use one format.
Figure 2-5 includes asterisks, the asterisks are not described in figure legends.
Figure 7 is authors own creation for this particular manuscript or it is adopted from somewhere else. If it is adopted then permission from the publisher is required.